# California Sea Cucumber (*Apostichopus californicus*) Abundance and Movement on a Commercial Shellfish Aquaculture Farm

Daniel L. Curtis [1], Christopher M. Pearce [1,2,*], Paul van Dam-Bates [1,3], Nicholas M. T. Duprey [1], Stephen F. Cross [2] and Laura L. E. Cowen [3]

[1] Pacific Biological Station, Fisheries and Oceans Canada, Nanaimo, BC V9T 6N7, Canada; dan.curtis@dfo-mpo.gc.ca (D.L.C.)
[2] Department of Geography, University of Victoria, Victoria, BC V8P 5C2, Canada
[3] Department of Mathematics and Statistics, University of Victoria, Victoria, BC V8W 3R4, Canada
[*] Correspondence: chris.pearce@dfo-mpo.gc.ca; Tel.: +1-250-756-3352; Fax: +1-250-756-7053

**Abstract:** California sea cucumbers (*Apostichopus californicus*) are often abundant at oyster farms in British Columbia, Canada both on the suspended gear as juveniles and on the seafloor beneath them as a mixture of juveniles and adults. Their natural abundance, high value, and potential to mitigate benthic organic loading has led to an interest in their coculture with oysters. Whether farmed sea cucumbers ought to be contained to physically separate them from wild stocks is debated. The present three-year field study examined the movement of wild California sea cucumbers on/off an operational oyster farm (~3000 m$^2$) to help inform future sea cucumber aquaculture development. Sea cucumber effects on organic loading, immigration to/emigration from the farm, and the efficacy of various containment-material mesh types and sizes were examined. Juvenile and adult sea cucumber densities on the farm steadily increased from the end of winter through the end of summer, likely due in large part to juveniles falling off the suspended oyster gear, which occurred at an average rate of ~780 ind d$^{-1}$ (for the whole farm) in the summer months. The largest increase in abundance on the farm was observed between January and March/April, when the population increased by 100–350 ind d$^{-1}$. Between late summer and early winter, sea cucumbers emigrated from the farm at a rate of 50–90 ind d$^{-1}$, neither juvenile nor adult densities on the farm changing appreciably over the winter. The sea cucumber density showed a progressive decrease in the first 20 m from the farm, after which the animals were scarcely noticed. *Apostichopus californicus* did not significantly decrease sediment organics beneath the farm compared to a nearby control site, but such an effect may have been lost due to their seasonal feeding cycles and/or the presence of other benthic grazers that were not part of our exclusion trial. Overall, our findings suggest that the separation of farmed and wild California sea cucumbers on a shellfish farm can only be guaranteed through containment, given the dynamic immigration and emigration patterns of wild stocks. Through laboratory trials, we found that individuals of *A. californicus* were able to squeeze through mesh as small as 32% of their contracted width and could escape fenced areas (90 ± 4% escape from nylon fencing and 40 ± 8% escape from Vexar$^{TM}$ fencing) unless the fencing extended above the water surface (where there was no escape from either type).

**Keywords:** dive survey; *Holothuroid*; movement; ocean ranching; sedimentation





## 1. Introduction

Aquaculture of the Pacific oyster (*Crassostrea gigas*) in British Columbia (BC), Canada is a highly valuable industry, 7,997,000 kg, worth CAD 16,044,000, being produced in 2021 (DFO statistics: https://www.dfo-mpo.gc.ca/stats/aqua/aqua20-eng.htm, accessed on 3 August 2023). Wild fishery of the California sea cucumber (*Apostichopus* (*Parastichopus*) *californicus*) in BC is also a lucrative industry, 605,546 kg worth about CAD 12,300,000 being

fished in the same year [1]. Presently, though, there is no commercial sea cucumber aquaculture industry in BC. However, recent increases in the price of wild-caught California sea cucumbers, along with worldwide concern for the conservation of many species of highly valuable tropical sea cucumbers [2] have led to increased interest in the culture and/or sea ranching of this species in BC and Washington, USA [3–5]. Interest is primarily driven by shellfish and finfish aquaculture proponents who are thinking of farming California sea cucumbers at existing suspended shellfish culture sites or finfish net-pen sites. That concept fits well with the principles of integrated multitrophic aquaculture (IMTA) [6], whereby cocultured deposit feeders (e.g., sea cucumbers) may help to ameliorate increased organic loading associated with aquaculture via their feeding activity, which recycles nutrients and bioturbates sediments [7–11]. The first objective of this study was therefore to determine if high densities of California sea cucumbers on shellfish aquaculture sites were able to fully or partially mitigate the increased organic loading associated with farming activities.

While previous studies have suggested that the use of sea cucumbers may be an effective means of mitigating organic loading at aquaculture sites (e.g., [3–5,7,8,11]), coculture with such deposit feeders is not without concern. Some proponents believe that no fencing or caging is required to keep cultured sea cucumbers within the boundaries of the aquaculture tenure, the premise being that the animals have a vested interest in remaining in areas of high food concentration [12,13]. Even if emigration was minimal, however, there is also concern about the immigration of wild sea cucumbers that may be attracted to the increased sediment organic levels at aquaculture sites. For one species of sea cucumber, there is evidence that the mixing of cultured and wild stocks is negligible: Slater and Carton (2010) [12] observed a stable isotope signature in *Australostichopus mollis* collected beneath a mussel farm that was consistent with that of farm-impacted sediment and distinct from the stable isotope signature of *A. mollis* collected from natural reefs nearby. It was concluded that the shellfish farm site demonstrated a high sea cucumber retention and that negligible mixing occurred between sea cucumbers at the farm and those at natural reefs nearby [12].

Our work has shown that although California sea cucumbers are not attracted to sites of high organic content per se, if they randomly encounter such areas, they alter their foraging behaviour, which may serve to retain them there for an indefinite period [14]. Since the conditions of licence allow BC shellfish farmers to retain wild animals on aquaculture tenures when they are harvested along with cultivated individuals for which they are licensed [15]—and there is difficulty in discriminating between wild and cultured stocks—there is the potential for wild sea cucumbers to be harvested along with cultured individuals. That would ultimately reduce the number of sea cucumbers available to the wild fishery and increase the fishing pressure on the wild stock if these removals were not included in the wild fishery's total allowable catch (although sea cucumber larvae, produced by wild and/or cultured individuals, settling on oyster culture gear could eventually recruit into the benthos as juveniles, once they are dislodged from the gear, and bolster wild populations). The second objective of our study was therefore to determine if there was immigration or emigration of sea cucumbers to or from the area of increased organic loading associated with an active commercial shellfish aquaculture site.

Another reason for the increased interest in the culture of *A. californicus* at existing shellfish farms is the high densities of individuals that are often already found on the seafloor at those sites. The likely reason is the settling of larval sea cucumbers on the suspended shellfish (primarily Pacific oysters) and gear, which provide a highly complex matrix where juveniles are able to grow essentially free from most predators. Those juveniles can be knocked off the suspended shellfish by storms or during shellfish harvest events every 2–5 years. Once on the seafloor, shellfish aquaculture sites, which typically have a thick layer of shell hash, provide refuge for typically cryptic juvenile California sea cucumbers [16] and create a hard substrate where they can selectively feed on high-organic-content faeces and pseudofaeces deposited by the shellfish [17]. That provides a unique mechanism whereby shellfish farms are essentially "self-seeding" for California sea cucumbers. To appreciate the impact aquaculture may have on wild sea cucumbers,

it is important to understand the potential contribution of those wild-set individuals to the population on/off the shellfish farms. The third objective of this study was therefore to determine the approximate rate at which wild-set juvenile *A. californicus* drop from suspended cultured shellfish/gear onto the seafloor.

If sea cucumber immigration and emigration occur on shellfish farms, it will be necessary to determine effective means of containment. That is particularly pertinent given the logistical difficulties associated with attempting to contain a large-size range of animals that can scale on vertical or inverted surfaces and whose only prominent hard part is their calcareous oral ring [18]. Therefore, the fourth and final objective of our study was to determine, through a series of laboratory experiments, if California sea cucumbers could be contained using fencing, rather than sealed cages, and what mesh types and sizes were effective at doing so.

## 2. Materials and Methods

*2.1. Field Movement Experiments*

2.1.1. Experimental Site

Field work was carried out at a Pacific oyster farm in Village Bay, Quadra Island, BC (50°9.373′ N, 125°11.602′ W). Village Bay has a wide mouth located on the southeast side of the bay, which opens to the Strait of Georgia (Figure 1). The bay is a maximum 37 m deep in the centre with a shallow shoal (8.2 m) just to the south of the centre in the mouth of the bay. The site is subject to strong currents and significant tidal flushing [19]. The farm consisted of 36 oyster rafts (6 × 6 m) arranged in three rows, the total area covered by the oyster rafts being ~3000 m$^2$ (Figure 1). Each raft contained approximately 250 6 m long strings of oysters, with about 180 oysters per string, for a density of ~1077 ind m$^{-2}$.

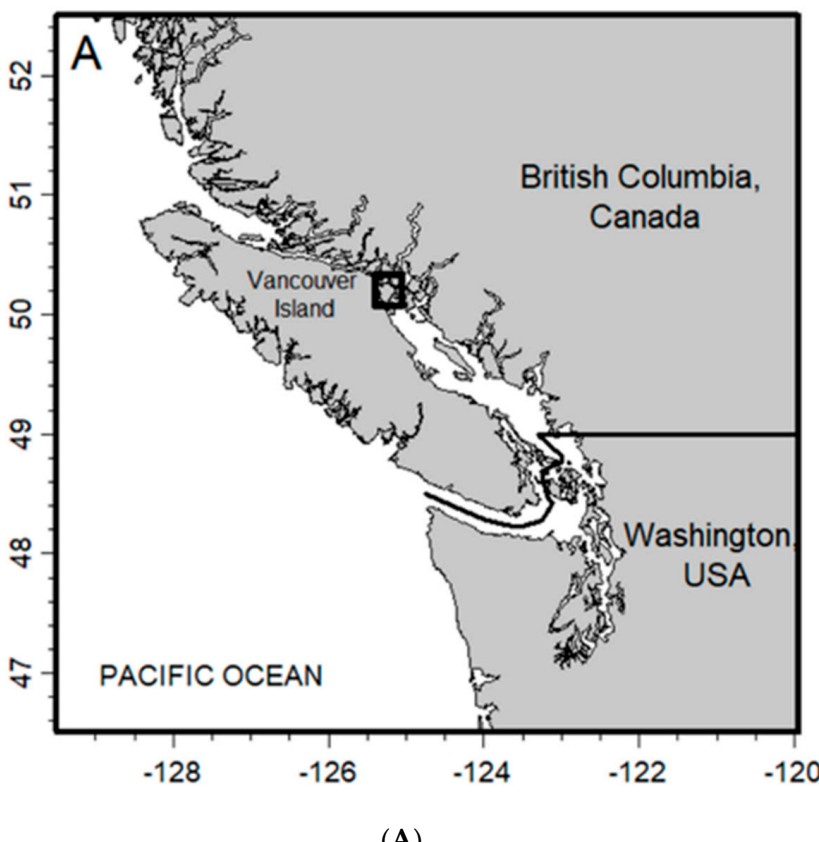

**(A)**

**Figure 1.** *Cont.*

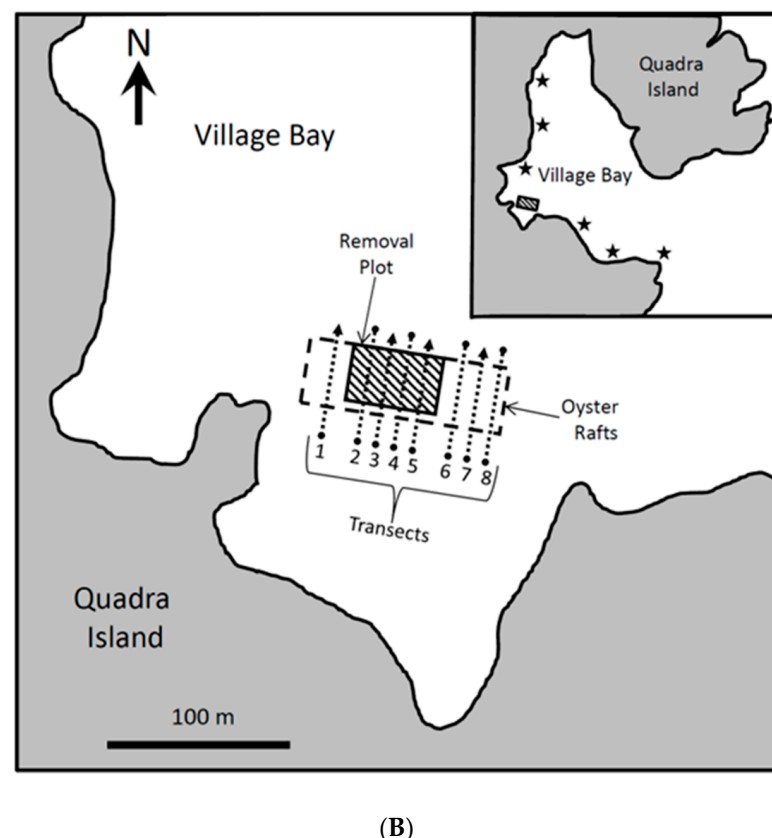

**(B)**

**Figure 1.** (**A**) Black-bordered box indicating the approximate location of the study region, off the east coast of Vancouver Island, British Columbia, Canada. (**B**) Location of the Viking Bay Ventures shellfish farm site in Village Bay, Quadra Island, British Columbia (50°9.373′ N, 125°11.602′ W). Long dashed lines indicate the perimeter of the farm. Short-dashed lines indicate the location of farm transects, those ending in an arrow indicating the deep farm transects. The cross-hatched rectangle indicates the perimeter of the sea cucumber removal plot. Stars in the insert box indicate the location of shoreline transects radiating out from the farm.

2.1.2. Experimental Design and Density Estimates

To provide an indication of the movements of adult sea cucumbers around the oyster farm, a removal plot was established within its area of influence (Figure 1B). All sea cucumbers were removed from an area measuring 50 × 30 m, simulating a harvest event. The removal plot, about 50% of the area of the farm, was delineated by permanent markers (cinder blocks) at each corner that were temporarily connected by a lead line during harvesting. Six permanent 60 m long transects were established, four through the removal plot (at 10 m interval spacing) and two 10 m outside the removal plot on either side, but within the area of impact of the shellfish gear (designated as "farm transects"). Those were originally surveyed in September 2012 immediately prior to the sea cucumber harvest. During the 2 d simulated harvest event, approximately 5000 sea cucumbers were collected from the removal plot, after which it was inspected, and any remaining sea cucumbers removed (those were all moved to a site ~1 km away from the farm). The removal plot was not harvested again during the study. The farm transects were resurveyed in January 2013. On subsequent sampling dates (April and August 2013, January, March, and July 2014, and January and March 2015), another two transects (making a total of eight farm transects) were added at 10 m spacing to the east of the plot, ensuring that the entire area of the farm was being surveyed (Figure 1B). Every other farm transect (i.e., transects 1, 3, 5, and 7; Figure 1B) was extended into deeper water away from the farm, and the portions of these transects extending into deeper water were treated separately and designated as "deep farm transects".

In an attempt to verify if sea cucumbers emigrated from the farm to the surrounding area or vice versa, in July 2014 and January and March 2015, an additional six transects were established along the shoreline, three on either side of the farm at approximately 200 m intervals, which were also surveyed (designated as "shoreline transects", Figure 1B). All transects were surveyed in accordance with established protocols used by Fisheries and Oceans Canada [20]. Briefly, each transect line was marked at 5 m intervals and surveyed by a pair of divers. The number of sea cucumbers was counted in a 2 m swath by one diver on the left and one diver on the right side of the transect line, effectively creating a series of adjacent 10 m$^2$ quadrats. Counts from the left and right divers were combined for density estimations. For farm transects, density was calculated as ind m$^{-2}$. For shoreline transects, density was calculated in two ways: (1) as ind m$^{-2}$, as is common in ecological studies, and (2) as ind (m of shoreline)$^{-1}$, as is used for sea cucumber stock assessment by Fisheries and Oceans Canada. The substrate type and algal cover were noted for each quadrat by the left diver. Farm transects were 60 m long. For deep farm transects, as the divers moved away from the farm, transects were surveyed until four quadrats with no sea cucumbers were observed or until a depth of 20 m (gauge depth) was reached, typically at a distance of 30 m. Shoreline transects were oriented perpendicular to the shore and surveyed from an 18 m gauge depth to the surface in accordance with Fisheries and Oceans Canada stock assessment protocols [20].

### 2.1.3. Sedimentation Rate

Deposited material was collected using sediment traps, beginning August 2013 and at each of five subsequent sample points until April 2015. Four replicate sediment traps were placed on the seafloor beneath the farm, while an additional four replicate sediment traps were set at a control site outside the area of influence of the farm [19]. Each sediment trap consisted of a PVC cylinder (diameter × height: 100 × 400 mm) with a cap on the bottom end, placed vertically, and anchored to the seafloor with rebar. The dimensions of the sediment traps were chosen to minimize the resuspension of trapped material, while maximizing the collection of suspended particulates in calm waters [21]. Sediment traps were deployed for 3–5 d to ensure that enough material was collected for the analysis and that bacterial degradation was minimal. The traps were deployed and collected by SCUBA divers. During collection, divers gently capped the tops of the sediment traps and brought them to the surface where they were kept cool until they could be further processed and frozen. Prior to freezing, samples were allowed to settle for ~30 min, after which excess water was gently decanted and the remaining sediments rinsed with fresh water into a smaller container and stored at −20 °C. Frozen sediments were thawed, decanted, and rinsed with fresh water into 50 mL centrifuge tubes. They were then centrifuged (3000× $g$, 10 min, 4 °C), decanted, and dried to constant weight at 60 °C. The material was next weighed to determine the rate of sedimentation and a subsample sent to the Department of Earth, Ocean, and Atmospheric Sciences at the University of British Columbia for the determination of total organic carbon (TOC) and total nitrogen (TN) (see [4] for methods). That analysis allowed for the determination of the carbon-to-nitrogen (C:N) ratio, an indication of nutritional quality [22,23].

### 2.1.4. Benthic Sediments

Beginning April 2013 and at each of six subsequent sample points until April 2015, duplicate benthic sediment samples were taken from the same five quadrats within the removal plot and five outside the removal plot, all along the original six farm transects (see above for a description of the transect locations). The quadrats for sediment samples were initially randomly chosen, but the same quadrats were repeatedly sampled at each subsequent sample point. Samples were taken from the top 5 cm of sediment using a 10 cc syringe with the end cut off. Following collection, sediment samples were kept cool and immediately frozen (−20 °C) upon return to the laboratory. Prior to the analysis of total organic matter, samples were thawed, freeze-dried, and particles larger than 1 mm

removed by sieving. Samples were then dried to constant weight at 60 °C and allowed to cool in a desiccator. Total organic matter, reflecting the potential macronutrients in the sediments available to the sea cucumbers, was determined based on the ash-free dry weight following combustion at 500 °C for a minimum of 5 h.

### 2.1.5. Sea Cucumber Drop-off Rate

The number of juvenile sea cucumbers on 10 oyster strings from four different rafts (40 strings total) were counted by divers using SCUBA on 18 July 2014, the same strings being counted again on 26 August 2014. Although little information exists on the length at maturity of *A. californicus*, following previous studies, sea cucumbers less than 14 cm (the length of a diver's underwater pencil) in relaxed length were considered to be juveniles [20].

### 2.2. Laboratory Containment Experiments

### 2.2.1. Fence Height and Type

Two types of material were used to test the effectiveness of fencing as a barrier to sea cucumber movement: a stiffer 11 mm Vexar$^{TM}$ mesh (Complex Plastics Inc., Elkhart, IN, USA) and a more flexible 10 mm nylon net. Circular fences with a diameter of 2 m were placed in a 3 m diameter tank with 1 m of water. Two fence heights were tested, halfway to the surface (0.50 m) and just above the surface of the water (1.03 m), which were totally crossed with material type, leading to four treatments (i.e., Vexar$^{TM}$ fence at half height, Vexar$^{TM}$ fence at full height, nylon fence at half height, nylon fence at full height). We had access to only one tank, so one replicate of each of the four treatments was run at a time, with the order of the four treatments randomized within a temporal block. Each treatment within a block was run for 47 h, with 1 h between treatments for measurements, dismantling, and set-up. Six blocks were run across time (i.e., *n* = 6) and the experiment was conducted from March to June 2013.

A lead line was used to weigh the base of the fence against the tank floor and prevent the sea cucumbers going under. The nylon net was held up by a set of foam floats whereas the Vexar$^{TM}$ mesh needed no flotation. The only lighting was a small amount of ambient light that got through gaps in the tank lid. The sand-filtered and UV-sterilized water flow was 28 L min$^{-1}$ through a tee, which dispersed the flow evenly in both directions. At the beginning of each trial, temperature, salinity, and dissolved oxygen were measured, 10 adult sea cucumbers were placed in the centre of the fenced area, and the tank was covered. After 47 h, the number of animals that had escaped was recorded, the contracted length (CL; 202 ± 2 mm, N = 240) and contracted width (CW; 73 ± 1 mm, N = 240) of sea cucumbers were measured, and the size index (SI) (1.5 ± 0.0, N = 240) calculated. The contracted length and width were measured after 10 s of handling. The size index was calculated as SI = CL × CW × 0.01, following [24].

### 2.2.2. Mesh Size

In preliminary containment trials, adult animals were observed squeezing through the mesh of wire cages. Optimal fencing should use the largest sized mesh possible to ensure good water flow while still preventing escapement. Six sizes of Vexar$^{TM}$ mesh and four size classes of California sea cucumbers were tested in this experiment in a totally crossed experimental design (i.e., 24 treatments). The mesh sizes were 4, 7, 11, 15, 21, and 38 mm (measured as the length of a mesh-opening side). Four size classes of sea cucumbers were tested—"Small Juveniles", "Large Juveniles", "Small Adults", and "Large Adults" (Table 1)—with only one animal of each size being present in an enclosure at a time. Enclosures were 0.5 m in diameter and 0.33 m high and extended above the water surface. Trials were run in six separate tanks (L × W × H: 1.22 × 0.92 × 0.30 m), with only one enclosure per tank and one replicate of each mesh size being run at the same time. Six blocks were run across time with each block having one replicate of each of the six mesh-size treatments, the placement of which was a split-plot design (i.e., each mesh-size treatment being run in each tank across time). Each temporal block was run for 47 h back

to back, with 1 h in between for measurements, and the experiment was conducted in June 2013. Trials were monitored every 2 h between 0900 and 2000 h to ensure there was no exit from and re-entry into the enclosures. At the end of each trial, the number of sea cucumbers that had escaped was recorded. Water was sand-filtered and UV-sterilized, the flow maintained at 5 L min$^{-1}$, and the depth was 0.25 m. A photoperiod of 16:8 (h light: h dark) was established by illumination provided by overhead fluorescent lights.

**Table 1.** Category and size of sea cucumbers used in the mesh size experiment. Contracted width (CW), contracted length (CL), and size index (SI) are given for each size category.

| Large Adult | Mean | SE | Minimum | Maximum |
|---|---|---|---|---|
| CW (mm) | 82 | 2 | 58 | 108 |
| CL (mm) | 215 | 4 | 160 | 270 |
| SI | 1.75 | 0.04 | 1.33 | 2.38 |
| Small Adult | | | | |
| CW (mm) | 64 | 2 | 44 | 80 |
| CL (mm) | 177 | 5 | 130 | 240 |
| SI | 1.13 | 0.04 | 0.66 | 1.52 |
| Large Juvenile | | | | |
| CW (mm) | 28 | 1 | 14 | 37 |
| CL (mm) | 94 | 3 | 58 | 145 |
| SI | 0.27 | 0.01 | 0.10 | 0.39 |
| Small Juvenile | | | | |
| CW (mm) | 18 | 1 | 13 | 37 |
| CL (mm) | 71 | 3 | 48 | 128 |
| SI | 0.13 | 0.01 | 0.07 | 0.41 |

### 2.3. Statistical Analyses

The sediment deposition data (total deposition, carbon, nitrogen, C:N ratio) did not meet the normality and homogeneity of variances assumptions of an ANOVA so PERMANOVA models were used to assess the fixed effects of the site (two levels), date (six levels), and their interaction on each sediment variable [25]. If the fixed effects or interactions were significant, further PERMANOVAs were run to investigate pairwise comparisons [25]. All models, main and pairwise comparisons, were run with 9999 permutations for each sediment variable assessed. Many of the pairwise PERMANOVA models had few (35) unique permutations so *p*-values of all of the PERMANOVA models were produced through Monte Carlo sampling [25]. The effect of the date on organic content of sediment inside and outside the removal plot (two levels) was compared using a two-way repeated measures ANOVA. The effects of the date (seven levels) on the number of sea cucumbers observed and maximum distance from the farm at which they were observed were assessed using a one-way repeated measures ANOVA and a rank-based repeated measures ANOVA, respectively, as the latter did not meet the homogeneity of variances assumption of the ANOVA. The effect of the date (seven levels) on the density and depth of occurrence of sea cucumbers along transects sited away from the farm were each assessed using one-way repeated measures ANOVAs. Sea cucumber drop-off rate from 10 oyster strings was compared using a one-way repeated measures ANOVA. In the laboratory fencing experiment, the sea cucumber escape from two different fence types was compared using a logistic regression. In the mesh-size experiment, either a mesh size worked or it did not on a given sea cucumber size, so no statistics were necessary. Means are reported with $\pm$ standard error, with the level of significance being set at *p* < 0.05. The normality and homogeneity of variances for data used in the ANOVAs were assessed using residual analyses and Brown–Forsythe tests, respectively.

## 3. Results

### 3.1. Deposition Rate

#### 3.1.1. Total Deposition Rate

There were patterns in the total sediment deposition rate. Although there was no main effect of the site on the rate of material deposition ($F_{1,36}$ = 0.0001, $p$ = 0.98), the effect of the site differed among the dates ($F_{5,36}$ = 15.35, $p$ < 0.0001), as indicated by the significant interaction between site and date ($F_{5,36}$ = 9.51, $p$ < 0.0001) (Figure 2A). There was a significantly higher total deposition rate at the farm than at the control site in August 2013 ($p$ = 0.05) and March 2014 ($p$ < 0.05), but a significantly higher total deposition rate at the control site than at the farm site in January 2014 ($p$ < 0.001) and January 2015 ($p$ = 0.07).

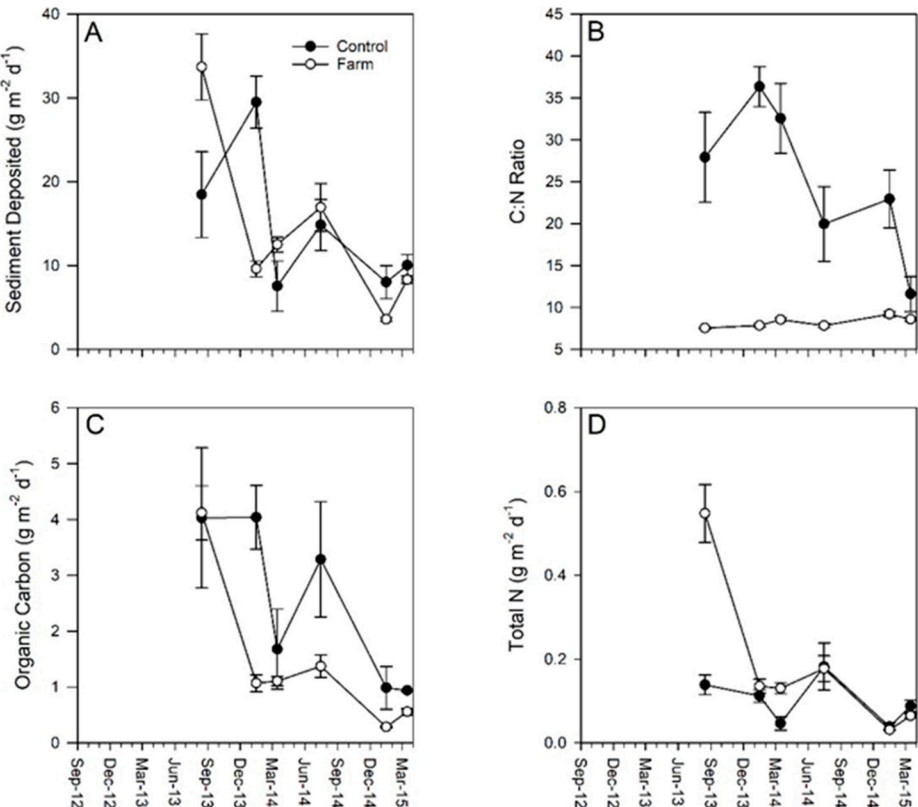

**Figure 2.** Mean (±SE) (**A**) total particulate matter sedimentation rate, (**B**) carbon:nitrogen ratio (C:N), (**C**) total organic carbon content, and (**D**) total nitrogen content of material deposited beneath the oyster rafts in Village Bay (solid symbols) and at the control site (empty symbols) outside of the area of influence of the farm ($n$ = 4). No trend lines were fit.

#### 3.1.2. C:N Ratio

The C:N ratio of the material deposited showed significant effects of the site ($F_{1,36}$ = 180.65, $p$ < 0.0001), date ($F_{5,36}$ = 8.68, $p$ < 0.0001), and the interaction between site and date ($F_{5,36}$ = 7.15, $p$ < 0.001) (Figure 2B). The C:N ratio at the control site was significantly ($p$ < 0.05) higher than at the farm at every time point. At both sites, there were significant seasonal differences in the C:N ratio. The C:N ratio at the control site tended to be higher in the winter and lower in the summer, a pattern that was obscured by a general decline over the course of the study. At the farm site, the C:N ratio also tended to be higher in winter and lower in the summer but increased over the course of the study. When comparing those trends, it is important to note the magnitude of changes in the C:N ratio between the sites. Changes in the C:N ratio at the farm were small, ranging from 7.54 ± 0.11 to 9.20 ± 0.33, whereas changes in the C:N ratio at the control site were more pronounced, ranging from 11.60 ± 2.09 to 36.36 ± 2.39. The level of carbonate carbon was higher at the farm than at the control site, indicating that there was more woody debris at

the latter, and/or that the former had more shell material (likelihoods both supported by visual observations).

### 3.1.3. Total Organic Carbon Deposition Rate

The rate of TOC deposition showed significant effects of both site ($F_{1,36}$ = 11.27, $p < 0.005$) and date ($F_{5,36}$ = 8.52, $p < 0.0001$), but there was no significant interaction (Figure 2C). The control site had a significantly ($p < 0.05$) higher TOC deposition rate than the farm site and there was a general trend towards a decreasing TOC deposition rate over time at both sites.

### 3.1.4. Total Nitrogen Deposition Rate

There were significant effects of the site ($F_{1,36}$ = 25.99, $p < 0.0001$) and date ($F_{5,36}$ = 22.25, $p < 0.0001$), as well as a significant interaction between the two factors ($F_{5,36}$ = 12.60, $p < 0.0001$) for the TN deposition rate (Figure 2D). Within the farm site, the TN deposition rate was significantly ($p < 0.05$) higher in August 2013 than at any other sample point. With the exceptions of a peak in TN deposition rate in August 2013 and following the start of the oyster harvest in February 2014, both the farm and the control sites showed similar patterns of TN deposition rate, lower values being observed in winter.

### 3.2. Benthic Sediments

There were no significant differences in the organic content of the sediments inside and outside the removal plot over the course of the experiment (Figure 3). That lack of a significant difference, however, may be due to a low statistical power. However, it is interesting to note that in March 2014, immediately following the beginning of the oyster harvest with a concomitant sharp increase in density of adult sea cucumbers within the removal plot, there was a corresponding notable decrease in the sediment organic levels before the organic content returned to normal levels in July 2014.

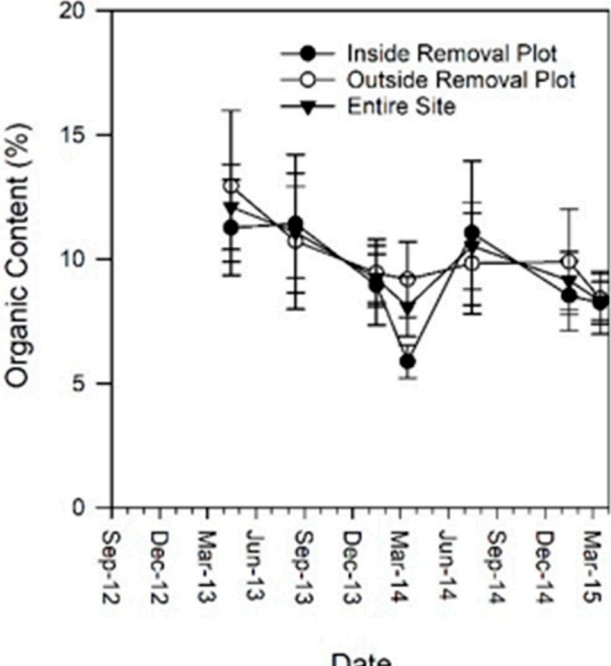

**Figure 3.** Mean ($\pm$SE) percent organic content of benthic sediments collected inside the removal plot (filled circles, $n = 5$), outside the removal plot (empty circles, $n = 5$), and pooled for the entire site (filled triangles) ($n = 10$). No trend lines were fit.

### 3.3. Density Estimates

#### 3.3.1. Adult Density

Prior to the harvest of sea cucumbers from the removal plot, densities of adult sea cucumbers inside and outside it were similar ($2.34 \pm 0.22$ and $2.16 \pm 0.21$ ind m$^{-2}$, respectively) (Figure 4A). In January 2013, about 4 months after the harvest of approximately 5000 sea cucumbers from the removal plot, the density of individuals in the removal plot remained low ($0.66 \pm 0.13$ ind m$^{-2}$) relative to the area outside it ($2.70 \pm 0.36$ ind m$^{-2}$). By March 2013, however, the density within the removal plot ($2.79 \pm 0.59$ ind m$^{-2}$) was similar to preharvest levels, and by August 2013, it was more than double ($5.98 \pm 0.59$ ind m$^{-2}$) that prior to harvest. Although the magnitudes of change in density were different following the return to pretreatment levels, the areas both within and outside the removal plot showed a similar pattern of change, increasing in spring/summer and decreasing in winter. That pattern was obscured by a general increase in density that was associated with the harvesting of the oysters. Maximum observed densities of $11.00 \pm 0.54$ and $6.67 \pm 0.46$ ind m$^{-2}$ of adult sea cucumbers inside and outside the removal plot, respectively, were observed in March 2015, at the end of the study, when nearly all of the oysters had been harvested.

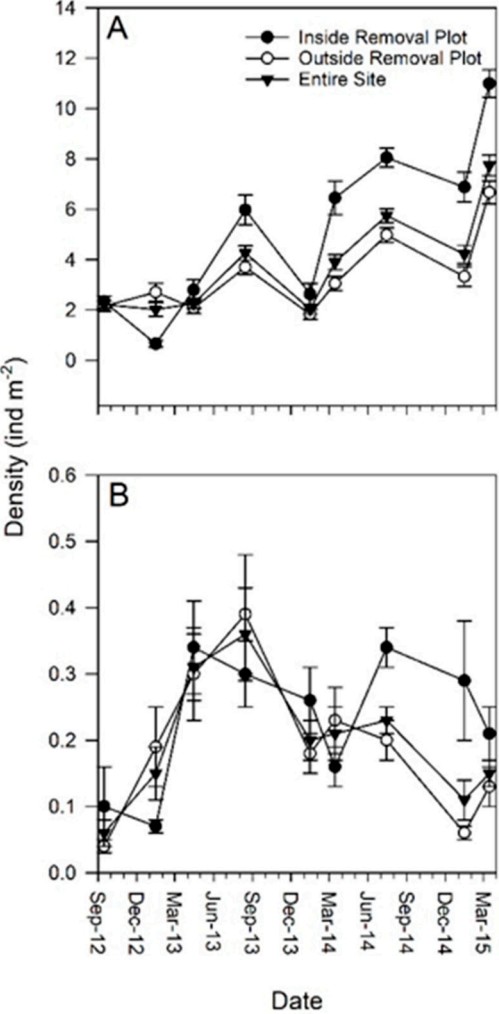

**Figure 4.** Mean ($\pm$SE) density of (**A**) adult and (**B**) juvenile sea cucumbers observed on the seafloor inside the removal plot (filled circles, *n* = 24), outside the removal plot (empty circles, *n* = 36 for September 2012 and January 2013 and *n* = 56 thereafter), and pooled for the entire site (filled triangles, *n* = 60 for September 2012 and January 2013 and *n* = 80 thereafter). Oyster harvest began in February 2014 and continued for the duration of the study. No trend lines were fit.

### 3.3.2. Juvenile Density

Although the juvenile density showed some patterns similar to that of the adult density, particularly in the removal plot, patterns in the former were less clear (Figure 4B). That is likely owing to their cryptic nature, an attribute that also contributed to observed densities that were over an order of magnitude lower than those of adults. Prior to the harvest, densities of juvenile sea cucumbers inside and outside the removal plot were similar ($0.10 \pm 0.06$ and $0.04 \pm 0.01$ ind m$^{-2}$, respectively). As with the adult sea cucumbers, following sea cucumber harvest, the density of juvenile sea cucumbers within the removal plot was less than that outside the plot. In contrast to the adults, however, that difference was the result of an increase in density outside the plot rather than a decrease within it. After the sea cucumber harvest in September 2012, the density of juvenile sea cucumbers throughout the site continued to increase until August 2013, reaching a maximum total density of $0.36 \pm 0.07$ ind m$^{-2}$. Over the winter and early spring of 2014, the density declined before increasing again later in the spring of that year, followed by a subsequent decrease. Unlike the adult scenario, the density of juveniles in 2014 did not exceed the maximum concentrations observed in 2013.

### 3.3.3. Deep Farm Transects

As distance from the farm site increased along the farm transects that extended from the farm into deeper water, sea cucumbers quickly became sparse, with similar densities to those beneath the farm occurring only within the first 10 m. There was a significant effect of the sampling date on the number of sea cucumbers observed ($F_{6,18} = 5.29$, $p < 0.01$) along transects that extended away from the farm towards deeper water (Figure 5A). The number observed in August 2013 was significantly higher than in April 2013, July 2014, and January 2015. Accordingly, there was a significant effect of the date on the maximum distance from the farm that sea cucumbers were observed ($X^2_6 = 15.39$, $p < 0.05$, $n = 28$), with the maximum distance noted in August 2013 being significantly greater than those in January 2015 and March 2015 ($p = 0.05$ and $p = 0.06$, respectively) (Figure 5B).

### 3.3.4. Shoreline Transects

When measured as either ind m$^{-2}$ or ind (m of shoreline)$^{-1}$, there were no significant differences in the density of sea cucumbers at shoreline transects located away from the farm site across sampling dates (Figure 6). There were also no significant differences in the average depth of occurrence across sampling dates for those that were observed on the transects.

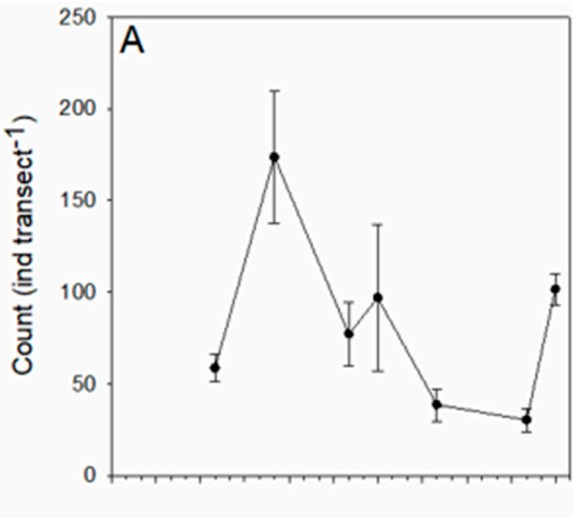

**Figure 5.** *Cont.*

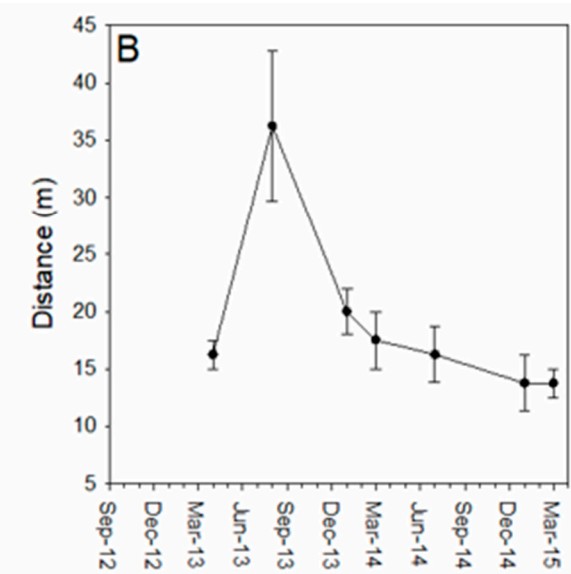

**Figure 5.** Mean (±SE) (**A**) number of adult sea cucumbers and (**B**) the furthest distance at which they were observed on the seafloor along transects extending away from the farm and into deeper water (transects 1, 3, 5, 7; *n* = 4). No trend lines were fit.

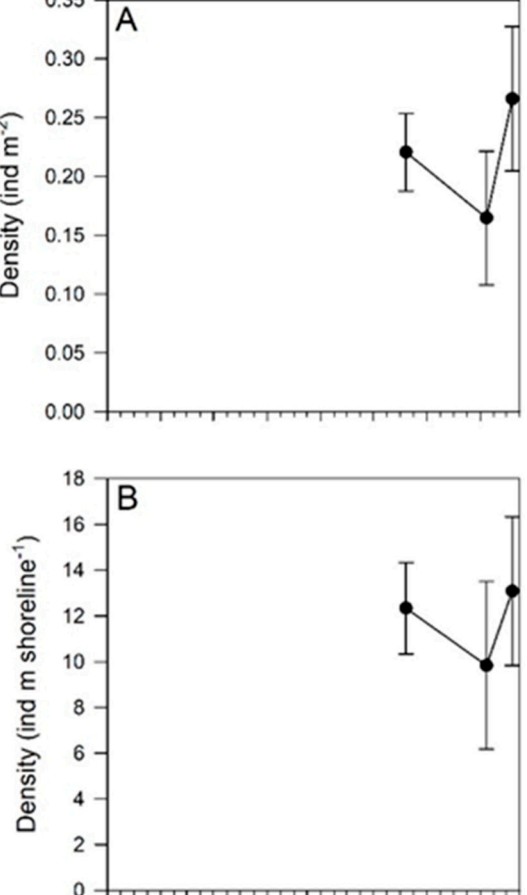

**Figure 6.** *Cont.*

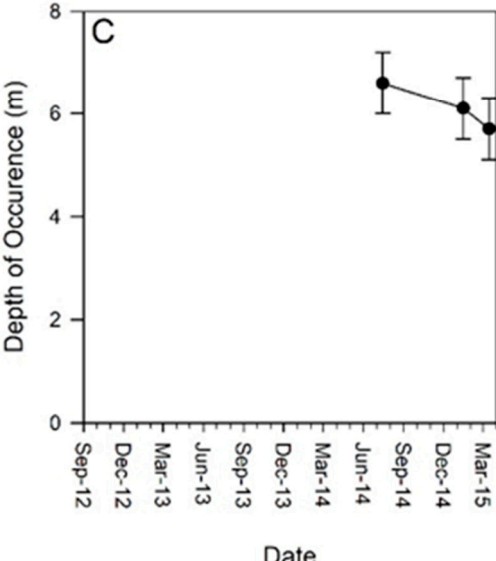

**Figure 6.** Mean (±SE) (**A**) density as ind m$^{-2}$, (**B**) density as ind (m shoreline)$^{-1}$, and (**C**) depth of occurrence (m) of adult sea cucumbers observed on transects radiating out along the shoreline at approximately 200 m intervals from the farm site within Village Bay (*n* = 6). No trend lines were fit.

3.3.5. Drop-off Rate

Over the period between July 18 and August 26, 2014 there was a significant decrease in the number of juvenile sea cucumbers observed on the oyster strings ($F_1$ = 30.46, $p$ < 0.001) (Figure 7). During that time, the density on the strings decreased from 10.70 ± 0.93 ind string$^{-1}$ to 7.51 ± 0.81 ind string$^{-1}$, a change equating to an average of 3.38 ind string$^{-1}$ for a drop-off rate of 0.087 ind string$^{-1}$ d$^{-1}$. With 250 strings per raft and 36 rafts on the farm, the entire site would experience a fall-off rate from the strings of ~780 ind d$^{-1}$, with ~30,380 ind for the entire 39 d period of observation.

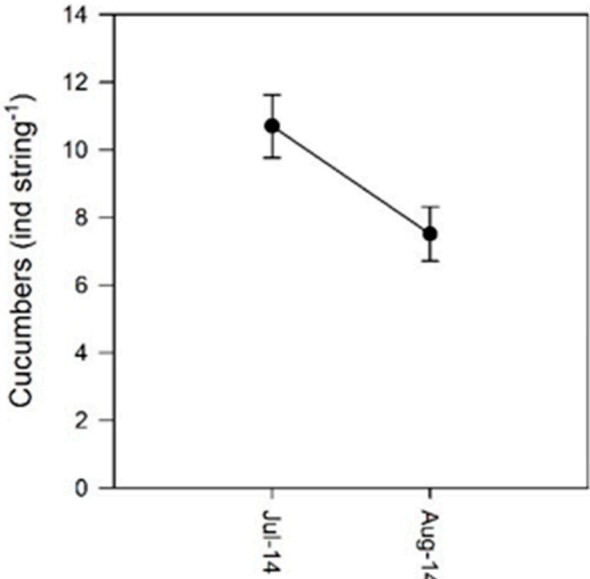

**Figure 7.** Number of juvenile sea cucumbers observed on oyster strings (*n* = 40) in July and August 2014. Data are mean ± standard error for the same strings measured at each sample point. No trend line was fit.

### 3.4. Laboratory Experiments

#### 3.4.1. Fence Type and Height

Mean ± SE temperature, salinity, and dissolved oxygen across all six temporal replicates were 10.6 ± 0.2 °C, 30.8 ± 0.1, and 10.4 ± 0.07 mg L$^{-1}$, respectively (N = 24). Out of 120 animals tested, there were no escapes from either fence type when the fences extended above the surface of the water. With regard to fences that did not reach the surface, significantly more animals escaped from the nylon netting (90 ± 4%) than from the Vexar$^{TM}$ mesh (40 ± 8%) (logistic regression estimated effect ± SE: 2.6 ± 0.5, $p < 0.001$).

#### 3.4.2. Mesh Size

There were no escapes of any-sized sea cucumber from the two smallest mesh sizes (4 and 7 mm) (Table 2). As the size of the mesh was increased by one level (11, 15, 21, 38 mm) an increasing size class ("small juvenile", "large juvenile", "small adult", "large adult") of sea cucumber was able to escape. Small juveniles escaped from mesh as small as 11 mm and large adults were able to escape from a mesh as small as 38 mm (Table 2). The smallest relative mesh size that cucumbers were able to escape from was observed for small adults, which were able to escape from a mesh that was 32% of their contracted width.

**Table 2.** Mesh size and the largest contracted width (max CW) of sea cucumber that was able to escape from each size of mesh (a given mesh size should effectively contain all sea cucumbers with a CW greater than the max CW).

| Animal Size | Mesh Size (mm) | Max CW (mm) |
|---|---|---|
| Large adult | 38 | 96 |
| Small adult | 21 | 64 |
| Large juvenile | 15 | 33 |
| Small juvenile | 11 | 20 |

## 4. Discussion

During times of high primary productivity and when high densities of oysters were stocked at the farm (August 2013 and March 2014), the farm site showed higher total sediment deposition rates than the control site, findings in harmony with those of [4] who found that peak sedimentation rates at the same farm site in 2004 occurred in April and July. Similar results were also shown by [26], who found that the contribution of biodeposits from mussels (*Mytilus edulis*) to the total sediment deposition was greatest during times of high primary productivity. In the current study, despite differences in the total sedimentation rate between the farm and the control site during times of high primary productivity (spring and summer, when total sediment deposition was higher at the farm site), the TOC deposition rate at both sites was similar. That result was due to higher levels of carbonate carbon deposition at the farm site, likely from deposited shell debris. Sedimentary material being deposited at the farm site showed a consistently lower C:N ratio than that deposited at the control site throughout the study, indicating that the former was of higher nutritional value than the latter (see [22,23]). Thus, although there was no additional organic carbon being deposited at the farm site during periods of high productivity, the material deposited was of higher nutritional value. Previous work has shown that the nutritional quality of available food as well as the volume of food available both play an important role in the growth of deposit-feeding sea cucumbers [27]. In addition, the hard substrate created by the layer of shell hash beneath the oyster farm likely facilitates selective feeding on such high-nutritional-quality biodeposits and provides cryptic spaces for juveniles to hide in. Therefore, it may be the quality and availability of food that allows some oyster farms to support high densities of sea cucumbers rather than the total amount of sediment deposition. In January 2014 and January 2015, the total sediment deposition rate at the control site was higher than at the farm. During those times, the TOC deposition rate was also higher at the control site than at the farm, but there was no

significant difference in the TN deposition rate, resulting in higher C:N ratios at the former. That indicates that there was an abundance of low-nutritional-quality organic material being deposited at the control site. Inspection confirmed that the material collected in the sediment traps at the control site contained a high proportion of fine woody debris. That was likely benthic material that was resuspended during storm events. An extensive study of Village Bay by [19] had similar findings during winter months. In that investigation, the benthic sediments in the area near the control site had a high proportion of woody debris that became more prominent in sediment traps during the winter.

Numerous benthic processes contribute to the breakdown and trophic transfer of deposited organic material. Although not as dramatic as at finfish farms, if left unchecked, the deposition of organic material beneath shellfish farms can eventually result in an anoxic state dominated by bacterial mats (e.g., [28]). The effects of epibenthic grazers on sediment organic content was qualitatively, and to some extent quantitatively, evident beneath the oyster farm at Village Bay. When all epibenthic grazers were completely excluded from small areas of the benthos, mats of *Beggiatoa* (a bacterium associated with high levels of organic loading and anoxia) quickly formed. When grazers were allowed back, the mats disappeared (Curtis, unpublished observations). That extreme shift was not observed, however, when California sea cucumbers were the only epibenthic grazers removed. If California sea cucumbers mitigate the accumulation of organic material beneath the farm, it would be expected that the organic content of the sediment would be linked to the density of sea cucumbers. Accordingly, there was a weak trend towards increased densities of California sea cucumbers and a decreased organic content of the sediment over the course of the study, but the decrease in organic content was not significant. Additionally, there was a sharp decrease in sediment organic content within the removal plot in March 2014 that was associated with a sharp increase in California sea cucumber density. One possible explanation for such a phenomenon is that changes in density were linked to seasonal changes in sediment deposition rate: during the winter months when the TOC deposition rate was low, the California sea cucumber density was also relatively low, and during the spring and summer, when the TOC deposition rate was high, their density was also higher. Further complicating that scenario is the fact that California sea cucumbers are likely preferentially feeding on biodeposits that settle on the hard substrate created by the shell hash beneath the farm and those materials are consumed or carried away by currents before they can become part of the organic content of the benthic sediment. Alternatively, a large number of green sea urchins (*Strongylocentrotus droebachiensis*) remained in the removal plot following the harvest of sea cucumbers, and previous work has shown that sea urchins actively consume biodeposits produced by aquaculture [29]. Therefore, such activity may have masked the relationship between the sediment organic content and California sea cucumber density when the sea cucumbers were removed. While the direct influence of sea cucumbers on sediment organic content is unclear, it is certain that epibenthic deposit feeders play a key role in mitigating the accumulation of organic matter in the sediments beneath shellfish farms. In future studies it may be possible to further refine the contribution of different members of the epibenthic community through more in-depth exclusion experiments.

The density of adult California sea cucumbers within the removal plot was slow to recover in the fall and winter months following their harvest. In addition, over the course of the study, the density of adult sea cucumbers beneath the shellfish farm showed a clear seasonal pattern that was overlaid by a general increase that was probably associated with the harvesting of oysters at the site. The failure of the sea cucumbers to redistribute beneath the farm between sea cucumber removal (September 2012) and when the area beneath the farm was first resurveyed (January 2013) is consistent with other studies that suggest that California sea cucumbers show little movement in the fall and winter (e.g., [30]). During that time, they resorb their internal organs, and it has been suggested that they enter a state of hibernation [4,30,31]. The fact that adult sea cucumbers moved less within the site during that time, combined with no change in overall density, also makes it unlikely that

they were moving off the site. Between winter and late summer during each year of the study, there was a net increase in the density of adult sea cucumbers in all areas of the study site. That increase was most pronounced between winter and spring, followed by a more gradual increase over the summer. Although those increases are likely the result of juvenile sea cucumbers falling from the oyster strings, being retained at the site, and growing up to become adults, the potential contribution of sea cucumbers moving onto the site from the surrounding area is unclear. Work by [12] on *Australostichopus mollis* in New Zealand, using stable isotopes to analyze their diet, suggests that sea cucumbers beneath shellfish aquaculture sites show a high site fidelity with minimal movement away from the site; however, the authors were unable to demonstrate that sea cucumbers from the surrounding area were moving into the site. In the current study, we were unable to show that sea cucumbers were retained at the site. Between late summer and early winter in the second and third years of the study, there was a clear decline in the density of sea cucumbers beneath the shellfish farm, with approximately 90 and 50 sea cucumbers per day leaving the site in 2013 and 2014, respectively. That decline was not evident in the first year. However, in 2012, sampling occurred in early September, which was later than in following years. Given the lack of dispersal within the site in the first year, it is probable that movements away from it occur in late summer/early fall, rates of movement being high during late summer/early fall and minimal thereafter. The window for observing movements away from the site may, therefore, have been missed in 2012.

Laboratory work has shown that the organic content of the sediment (and therefore available food) can alter sea cucumber foraging behaviour [14]. When the sea cucumber density is low and/or the food availability is high, they display a random foraging pattern. When resources become scarce, however, they display directed movements. In the current study, decreases in adult sea cucumber density were associated with decreases in total sediment, TOC, and TN deposition rates. Those changes were not significantly reflected in the sediment organic content, but that may have been masked by selective feeding on the hard substrate created by the layer of shell hash beneath the farm [4]. If our laboratory observations are applicable to a field setting, one possible explanation for the decrease in adult density is that in the late summer/early fall, sea cucumbers switch from a random foraging behaviour that would retain them at the site to more directed movements that could potentially lead them away from it. The number of sea cucumbers and the distance at which they were observed on the farm transects tended to be greater in August 2013 than during the other survey periods. The fact that a similar pattern was not observed at midsummer in July 2014 provides additional evidence that sea cucumbers may move away from the site in late summer/early fall (late August/early September). There were no clear patterns in the density of sea cucumbers in the shoreline transects located away from the farm, though that is not surprising given the amount of available habitat over which the sea cucumbers could disperse. Interestingly, despite anecdotal evidence that sea cucumbers seasonally migrate to deeper waters during winter, there was no clear seasonal change in the average depth of occurrence of individuals observed on the transects located away from the farm. While it is clear that California sea cucumbers move away from the farm site, likely in the late summer or early fall, it remains uncertain where they are going. It is also apparent that there is a net input of adults to the farm site in the spring and early summer. However, it was not possible to identify the relative contributions of movements onto the site and juveniles being knocked off from the overhanging oyster gear. Given the low density of adult California sea cucumbers in the surrounding area relative to the density of those at the farm site, it is likely that juveniles falling from the oyster strings were the main contributor.

In contrast to the density of adult sea cucumbers, the density of juveniles within the removal plot when first resurveyed (January 2013) was similar to preremoval values (September 2012), and the total density (inside and outside the removal plot combined) of juveniles more than doubled. A possible explanation is that very small and cryptic juvenile sea cucumbers on the seafloor had grown to such a size that they could be detected

in surveys or to a size where they could no longer be hidden within layers of shell hash. That is unlikely, however, since sea cucumbers show little growth during that time of the year and often actually get smaller [32]. A more plausible explanation is that the net input of juvenile sea cucumbers resulted from individuals being knocked off the oyster strings during fall and winter storms. When the density of juvenile sea cucumbers observed on the oyster strings (Figure 7) is extrapolated to the entire farm, it represents approximately 85,000 to 96,000 ind. During the summer months, we observed a drop-off rate of 0.087 ind string$^{-1}$ d$^{-1}$, which when extrapolated to the entire farm equates to approximately 780 ind d$^{-1}$ or 30,380 ind in the 39-day period of observation. Although this drop-off rate likely varies temporally with a variety of factors, the contribution of sea cucumbers recruited to the farm equipment/oysters to the benthic population could be sizeable. The rate of increase in density of adult sea cucumbers beneath the farm was highest in all years between the winter (January) and spring (either March or April) and was approximately 100, 180, and 350 ind d$^{-1}$ for 2013, 2014, and 2015, respectively. It is important to note that oyster harvesting began in February 2014 and continued for the duration of the study, the effects of which are unknown. Nevertheless, it is apparent that the contribution of juvenile sea cucumbers from the oyster strings due to the natural drop-off alone could easily account for the observed differences in adult density when oyster harvesting is not occurring. In terms of the total recruitment of sea cucumbers to the farm, one must not only consider the immigration of wild juveniles/adults and drop-off of juveniles recruited to the oyster strings, but also the natural settlement of wild larvae leading to juvenile recruitment. Pulses of recruits during "good" versus "bad" years for sea cucumber reproduction would likely have some impact on total recruitment within the farm.

The results of our study have shown that California sea cucumbers move away from the site and that they may also move onto it. Therefore, if there is a desire to prevent the mixing of wild and cultured individuals, some type of containment will be required. With regard to the fence height and type experiments, when fences did not extend to the surface of the water, more sea cucumbers escaped from fences made of nylon netting than from the stiffer plastic Vexar$^{TM}$ mesh. Although that seems counterintuitive, since one would assume that a stiffer substrate is easier to scale, there are a few possible explanations. The upper edge of the Vexar$^{TM}$ mesh creates a sharp edge, which may be an adverse stimulus. The nylon net is flexible and when sea cucumbers attempt to push themselves through, they distort the mesh, creating a larger effective opening, making it easier for them to escape. Lastly, in that and other experiments, we have observed California sea cucumbers displaying a strong thigmotaxis, whereby they press themselves against hard objects and minimize their movement; the stiffer Vexar$^{TM}$ mesh may have provided a better substrate for that behaviour. With regard to fences that extended above the surface of the water, there were no observed escapes from either type. That indicates that distortion of the nylon mesh is not the reason for the increased number of escapes since the sea cucumbers would be able to push themselves through the mesh at both fence heights and the maximum possible dimension of the collapsed netting was <20 mm. The lack of escapes when fencing reached the water's surface also indicates that sea cucumbers are unlikely to aerially expose themselves to scale over a fence. Although California sea cucumbers are occasionally observed in the intertidal, due to their reliance on hydrostatic pressure for locomotion, it is unlikely that they are able to move when out of the water. Therefore, in order to contain California sea cucumbers, it is probable that fences that protrude above the surface of the water or full enclosures will be required. The other issue with containing sea cucumbers relates to their lack of prominent hard parts [18] and subsequent ability to squeeze through small openings. In mesh-size experiments using a range of sizes, sea cucumbers were able to squeeze through stiff mesh ranging from 32 to 55% of their contracted width. Accordingly, it is recommended that mesh openings be no greater than 30% of the smallest sea cucumber's contracted width. In order to maintain optimal water and nutrient flow when culturing sea cucumbers, it may therefore be necessary to increase the size of the mesh used for containment structures as the sea cucumbers grow. The

size of mesh that would be required to ensure no movement of wild sea cucumber larvae (<1 mm) into enclosures would likely be too small to be used effectively in the field, as the very small mesh openings would likely become quickly constricted due to biofouling, hence restricting water flow. Thus, some wild larvae are likely to recruit into culture enclosures and be lost to the natural population and/or wild fishery. Conversely, adult sea cucumbers kept in enclosures are likely to spawn before they are harvested, and "cultured" larvae can cross the mesh barrier and be recruited back into the wild population.

The results of the present study have shown that through the deposition of faeces and pseudofaeces during times of high primary productivity, deep-water shellfish farms may provide a high-quality food source for deposit-feeding animals living beneath them. Additionally, the hard substrate created by shell debris beneath the farms may increase access to nutrients by facilitating selective feeding. In some cases, the combination of those factors likely helps to support higher densities of deposit feeders, which in turn may help to mitigate the increased organic deposition associated with shellfish farming. In our study, seasonal changes in the density of California sea cucumbers beneath the farm were observed, the density decreasing in the late summer/early fall and increasing in the spring/summer. Increases in the density resulted from either the addition of wild-set California sea cucumbers falling from the oyster strings and/or the immigration of adult sea cucumbers to the farm, while decreases in the density resulted from emigration of individuals away from the farm. These findings suggest that if shellfish farms are seeded with California sea cucumbers, some form of containment will be required to prevent the mixing of wild and cultured stocks. If the mixing of stocks is to be prevented, either full containment or fences reaching above the surface of the water will be required, the mesh size being determined by the size of the sea cucumbers to be enclosed.

**Author Contributions:** Conceptualization, C.M.P., D.L.C. and N.M.T.D.; methodology, C.M.P., D.L.C., N.M.T.D. and P.v.D.-B.; formal analysis, D.L.C. and P.v.D.-B.; investigation, D.L.C., N.M.T.D. and P.v.D.-B.; resources, C.M.P.; data curation, C.M.P., D.L.C. and P.v.D.-B.; writing—original draft preparation, D.L.C.; writing—review and editing, C.M.P., L.L.E.C., N.M.T.D., P.v.D.-B. and S.F.C.; visualization, D.L.C. and P.v.D.-B.; supervision, C.M.P., L.L.E.C. and S.F.C.; project administration, C.M.P.; funding acquisition, C.M.P. All authors have read and agreed to the published version of the manuscript.

**Funding:** This work was supported by Fisheries and Oceans Canada's Aquaculture Collaborative Research and Development Program (P-12-01-005, P-14-02-003), Fan Seafoods Ltd., Klahoose Shellfish Limited Partnership (Qathen Xwegus Management Corporation), and Viking Bay Ventures.

**Institutional Review Board Statement:** The animal study protocol was approved by the DFO Pacific Region Animal Care Committee of the Pacific Biological Station (protocol code 12-018 and date of approval 14 August 2012).

**Data Availability Statement:** Data are available upon request from the corresponding author (C.M.P.).

**Acknowledgments:** We thank the numerous people who contributed to the success of this study including Lyle Berzins, Dominique Bureau, Lyanne Curtis, Sarah Davies, Lindsay Dealy, Thomas Egli, Jessica Finney, Matt Grinnell, Joanne Lessard, Dan Leus, Janet Lochead, Jill Packham, and especially Seaton Taylor, for their assistance in the field; Troy Bouchard for sharing his knowledge, access to the site, and the use of his vessel; Lyanne Curtis for the statistical help; and Allie Byrne for assistance with the manuscript preparation.

**Conflicts of Interest:** The authors declare no conflict of interest.

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
