# Peer review of "California Sea Cucumber (Apostichopus californicus) Abundance and Movement on a Commercial Shellfish Aquaculture Farm"

_diversity, doi:10.3390/d15080940_

Round 1
Reviewer 1 Report
Review of manuscript #2452332 “California sea cucumber (Apostichopus californicus) abundance and movement on a commercial shellfish aquaculture farm” by Curtis, DL et al. submitted to Diversity.
General Comments
Strengths
1. Holothuroids are one of the lesser studied groups among echinoderms, they are commercially harvested, and have a strong impact on bioturbation. Multiculture systems are important, and this study is of practical application.
Weaknesses
1. Organization and some descriptions could be improved to provide more clarity, but easily revised.
Specific Comments
1. Introduction is good and to the point.
2. Order of methods 2.1 (pages 4-5)
2.1.2 Sedimentation rate - "August 2013 and at each subsequent sample point".
2.1.3 Benthic sediments - "April 2013 and subsequent sample point (eight)".
2.1.4 Experimental design and density estimates - this section has the details
It would be good to put the Experimental Design, with the description of time points, etc., before the two sedimentation sections, as reading it in the current order is a bit confusing.
3. Sedimentation rate 2.1.2 and Experimental Design 2.1.4 (page 4)
What/where is the control site? There are multiple different terms used throughout the manuscript - control, inside farm, outside farm, inside removal area, outside removal area, original transects, added transects, transects outside. A more structured description of when and where things took place would be good at the start of the methods. The map (Fig 1B) helps a lot, but the text description could be better organized.
4. Design description 2.2.1 fence height and type (pages 5-6)
"2x2 design" "One trial was run at a time within a single tank"
Four combinations for a 2x2:
One tank Vexar fence, half height
One tank Vexar fence, full height
One tank nylon fence, half height
One tank nylon fence, full height
Scenario A - 4 tanks available at one time
Four tanks setup at the same time, one for each combination, repeated 6 times
Scenario B - 1 tank available at one time
One tank setup for 4 different trial runs of 47 hrs one trial for each combination (="one" time point), repeated 6 times
I am assuming the experiment was scenario A, but it is not completely clear from the description.
5. Water measurements 2.2.1 (page 6)
2x2 = 4 combinations; repeated 6 times = 24 data points. Temperature & salinity have N = 24, which matches the design. Dissolved oxygen has an N = 56.
Where did the other 32 data points for DO come from?
6. Water & sea cucumber measurements Methods vs Results 2.2.1 (page 6)
Results are given with the Methods section, not the Results section. Methods section states temp, salinity, DO, CL, CW, SI were measured. Results section should give the outcome of those measurements. Not a huge issue, just one of format and presentation.
7. Mesh size design description 2.2.2 (page 6)
"six tanks, each tank 1.22 x 0.92 x 0.3 m. Six mesh sizes in each tank and enclosures were 0.5 m diameter". The numbers don't add up? Six enclosures at 0.5 m = 3 m long and a tank is only 1.22 m long. Side by side they would be at least 1 m wide and 1.5 m long and a tank is only 0.92 m wide, and 1.22 m long. Having cages really tightly packed together vs space between them may change escape response.
8. Table 1 Sea cucumber measurements 2.2.2 (page 6)
Table 1 details sizes. This might be better in the results, but it is part of the categorization for the methods. Either way, the copy that I received has a bunch of numbers 1., 2., 3. up to 80. throughout the whole table that are not measurements, which make no sense and make the table harder to read.
9. Design description 2.2.2 mesh size (page 6)
Six mesh sizes, and four sea cucumber sizes. As it reads now there are all sizes of sea cucumber inside one of each mesh size, which is a split-plot - all replicates of one factor (cuc size, split plot) inside one replicate of another factor (mesh size, whole plot blocked by tank). Latin-square is a cross-over design relative to split plot. "At the end of each trial, the number ..." - Was this (6 tanks at one time) repeated across multiple time points? What is a trial?
10. Statistical Analyses (page 7)
Analyses for the sediment, cucumber numbers, and fence type are given, but not the mesh size experiment.
11. Results
Most of the results sections are easy to follow, and concise. Table 2 (page 17) has the same issue as Table 1, with numbers 81., 82., up to 95. that are not measurements as part of the table.
The one thing I would point out, is that you really only have N = 1 for removal vs control as currently described in the methods and presented in the results. Subsampling one area multiple times does not produce replicates for testing removal (multiple nested measurement units are not replicate experimental units). If I wanted to compare male and female blood pressure, I would not measure one male 5 times at 3 different months and one female 5 times at 3 different months and "test" for a difference of male vs female with N = 3, or 5 or 15 - there is only one of each. I understand the limits of the field data having to use one farm, compared to the experimental lab data, but your qualitative description of changes (perfectly good and convincing) is stronger than the statistical tests in this case.
12. Discussion
The discussion is well written, easy to follow, and makes some good points. I would suggest a few things to think about in terms of the outcomes, especially relative to containment and mixing of wild vs farm stocks. 1. Across multiple years one would have to consider sea cucumbers settling on the bottom, not just on the strings or moving into the area at larger sizes. 2. Pulses of recruits during "good" vs "bad" years for sea cucumber reproduction would probably have some impact, and echinoderms are known for having good and bad recruitment years. 3. Perhaps most important for mixing, is larvae getting through any nets used for containment of juveniles/adults in both directions. Do sea cucumbers under farms spawn in the farm area?
Reviewer 2 Report
This paper focused on the abundance and movement of California sea cucumber on a commercial oyster aquaculture farm in British Columbia of Canada. The mass immigration and emigration of sea cucumbers in the oyster farm is an interesting phenomenon and this is a meaningful investigation. However, the experiment was conducted from 2013 to 2015. Till now, nearly ten years have passed, and the environments, resources, policy, and aquaculture might all change greatly. Furthermore, all the references in this manuscript are before 2017, yet new references should be cited. In addition, the information of table 1 and table 2 are confusing, and the authors should make all the information clearly. Some metrics such as the contracted width (CW), contracted length (CL), and size index (SI) are not commonly used, which should be explained why and how to calculate them.
Reviewer 3 Report
The present manuscript presents comprehensive and enough data set regarding ecology of Apostichopus californicus around aquaculturure farm. The authors conducted seasonal monitoring of distribution and density of A. californicus and compared them to organic matter in sinking particles and benthic sediments. The paper also contains data of drop rates of sea cucumbers from shellfish aquaculture string and those of escape from cages with various mesh sizes and structures. Those data are appropriately presented in this manuscript. I consider that those monitoring data are valuable for readers who have interest in integrated multitrophic aquaculture of sea cucumbers and shellfish, that is a hot topic in studies of invertebrate aquaculture.
This manuscript is generally well written and can be published almost as it is. I just suggest minor comments below.
Tables have unnecessary numbering in each item (e.g., "1. " Large Adult in Table 1).
Figure 5 (B): I could not understand what the furthest distance means and why the authors evaluated this parameter.
3.3.4 Away transects: I finally understand what they are, but recommend the authors to define the naming in the method section and uniformly use the naming throughout the text, to help understanding of readers.
3.3.5. Drop-off rate: Please discuss potential predation on juvenile sea cucumbers, as predated individuals did not contribute to the drop-off.
Round 2
Reviewer 2 Report
The authors should add the current situation of the oyster aquaculture industry and the sea cucumber resources in recent several years so as to help the understanding the scientific and practical meaning of the research nearly 10 years ago.
